# Comprehensive Analysis of Differentially Expressed CircRNAs in the Ovaries of Low- and High-Fertility Sheep

**DOI:** 10.3390/ani13020236

**Published:** 2023-01-09

**Authors:** Jinglei Wang, Hanying Chen, Yongsheng Zhang, Song Jiang, Xiancun Zeng, Hong Shen

**Affiliations:** 1College of Animal Science and Technology, Shihezi University, Shihezi 832003, China; 2School of Pharmacy, Shihezi University, Shihezi 832003, China

**Keywords:** cirRNA, RNA-seq, sheep, ovary, fecundity, ceRNA

## Abstract

**Simple Summary:**

Reproductive capacity is limited by the ovulation rate of ewes, which affects the sustainability of sheep farming. Recent studies have identified the involvement of circular RNAs (circRNAs) in follicular growth and development. Therefore, identification of circRNAs affecting follicle development could effectively improve the low number of prolific sheep breeds. In this study, circRNA expression profiles of follicular tissues of year-round estrus and prolific Cele black sheep, and seasonally estrous and low-yielding Hetian sheep were obtained using transcriptome sequencing technology. In addition, novel_circ_0040512 was determined to target sheep reproduction-associated oar-miR-125b using RNA pull-down and high-throughput sequencing techniques. These results provide a theoretical basis for the molecular biology of the genetic progression of sheep reproductive traits.

**Abstract:**

CircRNAs are essential in regulating follicle growth and development and the female reproductive system at multiple levels. However, the molecular mechanism by which circRNAs regulate reproduction in sheep is unclear and requires further exploration. In this study, RNA sequencing was performed to reveal the circRNA expression profiles in the ovaries of Cele black sheep and Hetian sheep during estrus. Analysis of the number of circRNAs in their host genes revealed that 5031 genes could produce 20,835 circRNAs. Among the differentially expressed circRNAs (DEcircRNA), 75 were upregulated, and 105 were downregulated. Functional enrichment analysis showed that the host genes of DEcircRNA were involved in several pathways, including the MAPK and Hippo signaling pathway. In addition, we constructed a subnetwork of competitive endogenous RNA (ceRNA) containing 4 mRNAs, 4 microRNAs (miRNAs), and 10 circRNAs, potentially related to follicle development. Functional circRNAs (e.g., novel_circ_0003851, novel_circ_0015526, novel_circ_0008117) were found to act as ceRNAs for follicle growth and development-related mRNAs (*CUEDC1*, *KPNB1*, *ZFPM2*) by sponging functional miRNAs (miR-29a, miR-29b, miR-17-5p). Finally, through an RNA pull-down assay, oar-miR-125b was selected and confirmed as the target miRNA of novel-circ-0041512. We analyzed the overall expression of circRNAs in sheep ovaries. Further, we explored the potential mechanisms underlying the circRNA functions, providing a theoretical basis for the genetic progress of reproductive traits in sheep.

## 1. Introduction

The profitability of high-fertility sheep farms affects the sustainability of the sheep industry. However, ovarian function and ovulation rate are important factors limiting sheep fertility. The ovulation rate in sheep can be affected by mutations in the genes *BMPR1B* and *BMP15* [1,2]. In recent years, great efforts have been made to uncover new genes (e.g., *SHC4*, *INHBA*, *ADCY7*, *CD44*, *PTGS2*, *CST6*, and *MEPE*) associated with reproductive capacity in sheep [3,4,5]. In addition, some miRNAs (oar-miR-125b, miR-99a, miR-150, and miR-27a) have been reported as important factors in sheep reproductive hormone biosynthesis and follicle development [6,7]. Ovine follicular development is a complex process that is regulated by multiple factors. To date, most studies on the mechanisms of follicular development have focused on candidate genes, transcription factors (TFs), and regulatory factors, such as miRNAs. These factors also interact, thereby forming a novel and variable functional regulatory network. Therefore, further identifying the factors affecting follicle development could effectively improve the number of prolific sheep breeds.

CircRNAs, a novel type of noncoding RNA (ncRNA), are formed by the reverse splicing of exons and introns to form a circular structure. CircRNAs are characterized by a covalently closed continuous loop and a canonical splicing junction site [8]. The unique circular structure of circRNAs distinguishes them from linear RNAs with 3’ and 5’ ends, thereby avoiding degradation by exonucleases [9]. Furthermore, circRNAs play important roles in the mammalian ovary [10], uterus [11], and pituitary [12]. In contrast, circRNA and miRNA contain numerous binding sites, which can act as miRNA “sponge” and enhance the target gene level by eliminating the inhibitory effect of miRNA. For instance, Circ EGFR targets miR-125a-3p, which helps control the Fyn gene to regulate follicular granulosa cell development [13]. In addition, the expression pattern of circRNAs is tissue-specific [14] and can affect gene transcription by binding to TF [15].

In recent years, the development of next-generation sequencing technologies has greatly facilitated our understanding of the type, quantity, and function of circRNAs in the reproductive organs of ewes. RNA-Seq was used to identify 10,226 circRNAs from embryonic and adult Kazakh sheep pituitary tissues [16]. A total of 183 DEcircRNAs were identified via expression profiling of follicular stage ovarian tissues from low- and high-fertility Humper sheep. These DEcircRNA host genes affected lambing numbers in sheep through the TGF-β and thyroid hormone signaling pathway [17]. CircRNA8073 in the endometrium of dairy goats can act as a ceRNA for miR-181a, thereby protecting neurotensin transcripts from miR-181a-mediated repression in endometrial epithelial cells (EECs) [18]. In addition, the sheep oviduct [14] and hypothalamus [19] have been increasingly reported to contain circRNAs.

Cele black sheep were developed through long-term crossbreeding selection between Kucha black lambs and other black lambs and exhibit excellent reproductive qualities, such as stable genetic performance and year-round estrus. In contrast, the semi-coarse-haired Hetian sheep in the same region have seasonal estrus and a single lamb in a single litter, with an average lambing rate of only 102.5%, which is lower than that of Cele black sheep (238%); the Hetian is a relatively low-product breed. Therefore, comparing the transcriptome levels of these two breeds through comprehensive analysis can help us identify the relevant genes and circRNAs, deepen our understanding of and provide a valuable resource for the molecular mechanisms related to reproductive traits in sheep.

## 2. Materials and Methods

### 2.1. Ethical Statement

Cele black and Hetian sheep were purchased from a sheep farm in Cele County, Hetian Region, Xinjiang Uygur Autonomous Region, China. All sheep were maintained under the same conditions and had free access to food and water. All experiments were conducted in accordance with the relevant guidelines and regulations established by the Ministry of Agriculture of the People’s Republic of China and were approved by the Animal Experiment Ethics Committee of the First Affiliated Hospital of Shihezi University School of Medicine (A2016-085).

### 2.2. Laboratory Animal Samples and Collection

First, 3 Cele black ewes (average number of lambs recorded per ewe = 3) and 3 Hetian ewes (average number of lambs recorded per ewe = 1) were selected based on their body size (35–40 kg) and age (3–4 years). Subsequently, all selected Cele black and Hetian sheep were synchronized to achieve estrus using vaginal sponges (injected with synthetic progesterone). The ewes were checked for estrus twice a day using test rams and a vaginal examination method. Finally, after detecting estrus, the ewes were euthanized using artificial anesthesia (xylene thiazoline, 1 mg/kg; Fei Long Animal Pharmaceutical Factory, Beian, China) by professional workers at a cattle and sheep slaughterhouse. Ovarian tissues were immediately collected and stored at −80 °C for RNA extraction. Cele black and Hetian sheep in estrus were named QE (n = 3) and HE (n = 3), respectively.

### 2.3. Total RNA Library Construction and Sequencing

Total tissue RNA was extracted using TRIzol reagent (Invitrogen™, Carlsbad, CA, USA), according to the manufacturer’s instructions. The extracted RNA was tested for quality and integrity (Agilent Technologies, Palo Alto, CA, USA). Subsequently, rRNA was removed from the total RNA using the EpicenterRibozero rRNA Removal Kit (EPIcenter, San Antonio, TX, USA), and rRNA-free residues were removed via ethanol precipitation. Sequencing libraries were generated using the NEBNext Ultra Directed RNA Library Preparation Kit (Illumina, San Diego, CA, USA). The library was sequenced, at Novogene (Beijing, China) using the Illumina HiSeq 4000 base platform, to generate 150 bp paired-end reads.

### 2.4. Raw Data Quality Control and Transcript Assembly

Poor-quality bases and adaptor sequences were removed from the raw data using the FastQC software (v0.11.9) to obtain clean reads. The reference genome index was constructed using bowtie2 (v2.2.8) and paired-end clean reads were aligned to the reference genome using HISAT2 (v2.2.1), followed by the reference sheep genome (Oar_v3.1). StringTie (http://ccb.jhu.edu/software/stringtie/index.shtml?t=manual; accessed on 13 December 2020) was used to assemble the mapped reads for each sample.

### 2.5. CircRNA Identification

The circRNAs were detected and identified using find_circ (v1.1) and CIRI2 (v1.2). If known circRNA data were available for this species, the known circRNAs identified were named after the circBase database name of the species. circRNAs without annotation were defined as novel discovery circRNAs. The Circos software was used to construct the Circos figure. The raw counts were normalized using TPM. Normalized expression level was calculated as: (readCount × 1,000,000)/libsize (where libsize is the sum of the circRNA read count).

### 2.6. Differential Expression Analysis

Differential expression analysis was performed using the DESeq R package (v1.10.1). DESeq uses a model based on a negative binomial distribution and provides statistical routines for identifying differential expression in numerical gene expression data. Benjamini and Hochberg’s method was used to control the false discovery rates, adjusted to obtain |log2(foldchange)| > 1 and *p*-value < 0.05 for circRNAs, miRNAs, and mRNAs designated as differentially expressed. A bubble-rank plot of the differential gene expression was plotted using the R ggplot2 package. Differential base-circRNA expression heat maps were plotted using the R pheatmap package (v1.0.12).

### 2.7. Functional Enrichment Analysis of CircRNAs

Based on the DEcircRNA host gene correspondence, gene ontology (GO) and Kyoto Encyclopedia of Genes and Genomes (KEGG) enrichment analyses were performed on the collections of host genes using the g:GOSt (https://biit.cs.ut.ee/gprofiler/gost; accessed on 13 September 2022) and KOBAS (http://http://kobas.cbi.pku.edu.cn/; accessed on 13 September 2022) online database, respectively, to explore the potential biological roles of the candidate genes. The results were visualized using the ggplot2 package of RStudio (v4.1.0).

### 2.8. Construction and TF Analysis for the Construction of Protein–Protein Interaction (PPI) Networks

To construct a PPI network, we searched the Search Tool for Interacting Genes/Proteins (STRING) database (https://string-db.org/; accessed on 18 September 2022). The network graph was visualized and analyzed using the MCODE plugin in Cytoscape (v3.7.1) for highly connected hub proteins throughout the network. AnimalTFDB (http://bioinfo.life.hust.edu.cn/AnimalTFDB/#!/; accessed on 23 September 2022) database was used for TF prediction.

### 2.9. Construction of DEcircRNA-DEmiRNA-DEmRNA Networks

CircRNAs are rich in miRNA binding sites, acting as miRNA sponges in the cell and alleviate the repressive effect of miRNAs on target genes. First, the prediction of circRNA-miRNA interactions was based on the miRanda software. Second, using the pre-transcriptome sequencing data of the project group [7,20], target gene prediction was performed on the known miRNAs obtained from the analysis by the intersection of the miRanda, PITA, and RNAhybrid software. The correspondence between miRNAs and target genes was obtained, the relationship between miRNAs and target genes was visualized using the Cytoscape (v3.7.1) software, and all relationship pairs with *p*-value < 0.05 were selected as potential DEcircRNA-DEmiRNA or DEmiRNA-DEmRNA pairs.

Based on the shared DEmiRNA binding sites and competitive relationships between DEcircRNAs and DEmRNsA, ceRNA pairs with a *p*-value < 0.05 were screened as the final ceRNA relationship pairs. Finally, the ceRNAs were visualized as an alluvial plot drawn using the ggalluvial package.

### 2.10. Ovarian Granulosa Cell Isolation and Culture

Sheep ovarian granulosa cells were collected in culture at a local slaughterhouse (Shihezi, Xinjiang Uygur Autonomous Region, China) during the breeding season. Briefly, collected ovarian granulosa cells were inoculated into Petri dishes supplemented with 10% fetal bovine serum (Gibco, Waltham, MA, USA) at 37 °C with 5% CO_2_. After the cells grew to (60–70%) density, the cells were collected for further experiments.

### 2.11. RNA Pull-Down Assays

Biotin-labeled oar-miR-125b probes (bio-oar-miR-125b-wt and bio-oar-miR-125b-mut) were synthesized by Cloud Sequence Biotechnology Ltd. (Shanghai, China); The probe sequence is shown in (Appendix A). RNA pull-down assays were performed with Pierce Magnetic RNA-Protein Pull-Down Kit (Thermo Fisher Scientific, Waltham, MA, USA) according to the instructions. In vitro transcribed (IVT) RNA probes for pull-down assays were prepared with AmpliScribe™ T7 High Yield Transcription Kit (Epicentre). In brief, 10^7^ cells were collected and wash in ice-cold phosphate-buffered saline. The cell pellets lysed in 1ml IP lysis buffer and centrifuged at 12,000 *g* for 15 min at 4 °C to collect the supernatant. Second, 50 μL washed streptavidin magnetic beads incubated with 5 μg biotinylated IVT lncRNA or its antisense RNA for 30 min at room temperature with agitation. Then, probes coated beads incubated with 500 μL cell lysys supernatant for 1 h. The beads were washed briefly with wash buffer for five times and elutioned. The bound protein or RNA in the pull-down materials were purified for further analysis.

### 2.12. RNA Library Preparation and Sequencing

CircRNA-Seq was performed by Cloud-Seq Biotech (Shanghai, China). Briefly, total RNA was pretreated to enrich circRNA using CircRNA Enrichment Kit (Cloud-seq, Inc., Shanghai, China). RNA libraries were constructed by using pretreated RNAs with NEBNext^®^ Ultra™ II Directional RNA Library Prep Kit (NEW ENGLAND BioLabs Inc., Ipswich, MA, USA) according to the manufacturer’s instructions. Libraries were controlled for quality and quantified using the BioAnalyzer 2100 system (Agilent Technologies, Inc., USA). Libraries were controlled for quality and quantified using the BioAnalyzer 2100 system (Agilent Technologies, Inc., USA). Library sequencing was performed on an Illumina Novaseq 6000 instrument with 150 bp paired-end reads.

### 2.13. RNA Pull-Down Data Analysis

Paired-end reads were harvested from the Illumina Novaseq 6000 sequencer, and were quality controlled by Q30. After 3′ adaptor-trimming and low-quality reads removing by cutadapt software (v1.9.3). The reads were aligned to the sheep reference genome (Oar v3.1) with bowtie2 software and circRNAs were detected and annotated with find-circ software. circBase database and circ2Trait disease database were used to annotate the identified circRNAs. The junction read counts were normalized and differentially expressed circRNAs were determined using edgeR package of R software. Corrected *p*-value < 0.05 was set as the threshold for significantly differential expression. GO and Pathway enrichment analysis were performed for the host genes of the differentially expressed circRNAs. CircRNA-miRNA interaction were predicted by targetscan and miranda softwares.

## 3. Results

### 3.1. Genomic Characteristics of CircRNA in Sheep Ovaries

A total of 222,290,378 (QE) and 227,351,666 (HE) raw reads were generated by sequencing the cDNA library using Illumina HiSeq 4000. After trimming adapters and removing low-quality reads, 217,171,270 (QE) and 220,967,234 (HE) reads were obtained. We observed an average of 10.86 G and 11.05 G clean bases and mapping rates of 96.63–97.98% (QE) and 97.49–97.67% (HE) (Table 1).

A total of 21,531 and 20,572 circRNAs were identified as QE and HE, respectively. circRNA expression profiling revealed more than 17,298 circular RNAs co-expressed in QE and HE (Figure 1A). Most identified circRNAs were 200–20,000 nucleotides (nt) in length (Figure 1B). Analysis of the number of circRNAs from their host genes revealed that one gene could produce multiple circRNAs and 20,835 circRNAs were produced from 5031 host genes (Figure 1C) (Appendix A). These reproduction-related candidate genes were mainly distributed on chromosomes 1, 2, and 3 (Appendix A). Based on the shear site information of circRNAs and relative position of the gene structure, more than 87%, 8%, and 5% of circRNAs were located in exon, intergenic, and intron regions, respectively (Figure 1D). In the normalized circRNA, the transcript expression levels of QE and HE were similar (Figure 1E).

### 3.2. Analysis of DEcircRNAs

The expression level of TPM was used to normalize the expression level of genes in each sample. CircRNAs with a *p*-value < 0.05 were considered to be differentially expressed. In total, 180 DEcircRNAs (75 upregulated and 105 downregulated) were further studied. Novel_circ_0005385 was the most significantly upregulated, whereas nov-el_circ_0005458 was the most significantly downregulated (Figure 2A,B) (Appendix A).

### 3.3. DEcircRNA Functional Enrichment Analysis

We performed GO and KEGG enrichment analyses of the DEcircRNA host genes to predict their potential functions. GO enrichment analysis showed that these genes were mainly involved in miRNA binding (GO:0035198), regulatory RNA binding (GO:0061980), miRNA-mediated gene silencing through inhibition of translation (GO:0035278), and cellular catabolic processes (GO:0044248) (Figure 3A). KEGG analysis identified genes enriched in pathways closely related to hormone secretion and cell proliferation, such as the MAPK signaling pathway (ID: oas04010), Hippo signaling pathway (ID: oas04390), and tight junctions (ID: oas04530) (Figure 3B).

### 3.4. PPI Network Analysis of Host Genes

To further elucidate the biological functions of DEcircRNA host genes, all DE-circRNA host genes in the two comparison groups were analyzed using the STRING database. After removing unlinked nodes, clusters of pivotal genes scoring above 0.1 in each PPI network were identified using the Cytoscape MCODE plugin to construct a PPI network consisting of 18 nodes and 18 edges (Figure 4). Subsequently, we sorted all the DE-circRNA host genes according to connectivity and score, novel_circ_0018428 (DOCK1) and novel_circ_0024392 (PPP2R2A), which contain the most nodes and scores, were considered as hub genes.

### 3.5. TFs Prediction and Analysis

CircRNAs are regulated by upstream TFs. Therefore, we aligned the promoter region sequences of putative hub genes (PPP2R2A and DOCK1) with the animal TFdb database for TF prediction. We identified 81 expressed TFs belonging to the 29TF family (Appendix A). A regulatory network of genes and TFs containing 70 nodes and 251 edges was constructed using the Cytoscape software. Our network analysis showed that ARRB1 binds directly to PPP2R2A and interacts with EP300 and SRC (Figure 5). Notably, PPP2R2A is an important regulator of proliferation and apoptosis in lambda endometrial stromal cells [21]. These results suggest that the TF ARRB1 may play an important role in follicle development in sheep.

### 3.6. Construction of a DEcircRNA-DEmiRNA-DEmRNA Regulatory Network

First, all DEcircRNA-targeting DEmiRNAs were predicted using the miRanda software. Among them, 263 nodes and 455 junctions were found between 180 DEcircRNAs and 83 DEmiRNAs. Notably, 11 DEcircRNAs (4 upregulated, 7 downregulated), which have been shown by the project team to be involved in sheep follicular development oar-miR-125b [7] targeting, were observed in this network (Appendix A, Appendix A). Potential DEmiRNA-targeting DEmRNAs were analyzed using miRanda, PITA, and RNAhybrid (Appendix A) software. Finally, based on the regulatory relationships of DEmiRNA-DEmRNA and DEmiRNA-DEcircRNA, four DEmiRNAs (one downregulated, three upregulated) were identified as DEcircRNA-predicted DEmiRNAs, and four DEmRNAs (three downregulated, one upregulated) were identified as DEmiRNA-predicted DEmRNAs (Figure 6) (Appendix A). Subsequently, KEGG enrichment analysis of the target genes was performed using the KOBAS 3.0 online database. The results showed that these target genes were involved in reproductive signaling pathways, such as the TGF-β signaling pathway, ECM receptor interaction, and prolactin signaling pathway (Appendix A).

### 3.7. Novel-circ-0040512 May Spongify oar-miR-125b in Sheep Ovarian Granulosa Cells

#### 3.7.1. RNA Pull-Down Calling oar-miR-125b-Binding CircRNA

CircRNAs mediate *STAT3* expression by regulating the miR-125b axis and affect its function [22]. Previously, our project team showed that oar-miRNA-125b expression was significantly upregulated in the ovaries of Cele black sheep compared to that in Hetian sheep and targeted the *STAT3* gene in ovarian granulosa cells [23]. In this study, a DEcircRNA-DEmiRNA network was constructed, and seven downregulated circRNAs that target oar-miR-125b were identified (novel_circ_0000458, novel_circ_0009713, nov-el_circ_0011620, novel_circ_0011939, novel_circ_ 0040508, novel_circ_0040512, and nov-el_circ_0019713). We observed a correlation between these DE-circRNAs and miRNA-125b expression. Accordingly, combined with high-throughput sequencing, miRNA pull-down was used to explore the oar-miR-125b binding of DEcircRNA. oar-miR-125b pull-down was first performed, followed by total RNA extraction for circRNA sequencing. The results showed that 3880 and 3666 circRNAs were identified in the biotin-labeled oar-miR-125b probe group (positive) and control group (negative), respectively (Appendix A).

#### 3.7.2. Functional Analysis of oar-miR-125b Potential Binding DEcircRNA

A total of 3040 DEcircRNAs were identified using edgeR and subjected to differential expression analysis, followed by a quasi-likelihood F-test to screen for DEcircRNAs (*p* < 0.05; Figure 7A,B) (Appendix A). These DEcircRNA host genes were subjected to GO and KEGG enrichment analyses. The results showed that they were enriched in BP, such as cellular localization, localization of cellular proteins, and localization of cellular macromolecules; CC, mainly belonging to the cytoplasm, organelles, and intracellular organelles with membranes; and MF, including ATP, adenosine nucleotide, adenine nucleotide, and enzyme binding (Figure 8A). The main KEGG signaling pathways included the MAPK, TGF-beta, and PI3K-Akt signaling pathways (Figure 8B).

Ultimately, we found novel_circ_0040512 in both DEcircRNA-DEmiRNA and RNA pull-down combined high-throughput sequencing results. In addition, novel_circ_0040512 was significantly upregulated in the oar-miR-125b probe group compared to that in the control group (Figure 9). circRNA subcellular localization is critical for its cellular function, and lncLocator 2.0 (http://www.csbio.sjtu.edu.cn/bioinf/ lncLocator/; accessed on 8 October 2022) was used to predict the subcellular localization of nov-el_circ_0040512. The results showed that novel_circ_0040512 was mainly distributed in the cytoplasm (Appendix A). Our results suggest that novel_circ_0040512 affects sheep follicle development by regulating the oar-miR-125b/*STAT3* axis; however, this finding needs further validation.

## 4. Discussion

In most mammals, the ovary is a gonad that exists in pairs. It can regulate follicular development through the cyclic production of gametes and reproductive hormones. The formation of primordial follicles by oocytes represents the first stage of follicular development. The progression of development and increase in follicle size leads to the production, maturation, and expulsion of the dominant follicle. Granulosa cells surround the entire follicle during this process, and their morphology and number change during the development of the follicle. Previously, our group showed that several miRNAs are involved in follicular development in Cele black and Hetian sheep [7].

So far, few studies have analyzed the circRNAs in sheep ovaries. A total of 4256 candidate circRNAs have been identified in single- and multi-parous Hanper sheep ovarian tissues [17]. A total of 3223 candidate circRNAs were identified, via analysis of the oviductal circRNA expression profiles of different FecB genotypes, in small-tailed cold sheep [14]. In the present study, 21,531 and 20,572 circRNAs were identified in the ovaries of Cele black and Hetian sheep, respectively, during estrus using transcriptome sequencing, thereby indicating that circRNAs are tissue-specific and developmentally specific at different stages [24,25]. In addition, 40 and 32 circRNAs were uniquely expressed in the ovarian tissues of Cele black and Hetian sheep, respectively, thus indicating strong breed-specific expression of circRNAs [25]. Furthermore, some circRNAs were expressed at higher levels than typical linear mRNAs; however, circRNAs normally accumulate at a low abundance [26]. In one of our studies, the ovarian transcriptomes of Cele black and Hetian sheep during estrus were analyzed using the same RNA samples as in the present study [20]. Some circRNA showed higher expression levels than the host genes. For instance, the expression level of novel_circ_0018981 was 227-fold higher than that of its host gene (*LAMA1*) in the ovarian tissue of Cele black sheep. In addition, novel_circ_0035676 and nov-el_circ_0019061 were 92-fold and 110-fold higher than the mRNA levels in Cele black sheep, respectively.

CircRNAs are derived from the same transcriptional products as the mRNAs of host genes and play important roles by positively or negatively regulating the transcription and expression of their parental genes [27,28]. Therefore, circRNAs are correlated with parental gene expression. We found that several known genes associated with follicle development produce multiple circRNA isoforms. For example, *COL14A1*, *TGFBR3*, and *SMAD2* has 33, 14, and 5 circRNA isoforms. Therefore, a single parental gene may produce several circRNA isoforms and, in some cases, more than 10. However, in general, only one or two isoforms were prominent, with low expression of most other isoforms, suggesting that circRNA cyclization is tightly regulated in the ovary. In addition, our study found that sheep had the highest number of host genes on chromosomes 1, 2, and 3 and the lowest number of genes on chromosome 26, suggesting that the distribution of circRNAs on chromosomes affects their host gene numbers, which is consistent with previous studies [17].

Based on the screening criteria of |log2(foldchange)| > 1 and *p*-value < 0.05, 180 DEcircRNAs were identified. For DEcircRNA host gene KEGG analysis, we observed that known follicle development-related pathways, such as the MAPK and Hippo signaling pathway, which are tightly linked to follicle growth and development, were enriched [29,30]. Genes enriched to a higher degree were *PARD3*, *PPP3CC*, and *PPP2R2A*. To further predict the potential role of differentially expressed circRNAs in sheep follicle development based on the PPI network and DEcircRNAs in the MCODE results. Notably, the *PPP2R2A* gene involved in the Hippo signaling pathway is highly linked to other nodes. *PPP2R2A* is essential for oocyte maturation and influences cell proliferation through regulation of and apoptosis affecting the physiological activities of sheep [21,31]. Since circRNAs can repress the transcription of host genes by binding to TFs [32]. Therefore, identifying potential TFs associated with circRNAs in host genes may reveal their functions. We found that the TF *ARRB1* targets the central host gene *PPP2R2A*. *ARRB1* is a protein-coding gene that regulates follicular development [33]. *ARRB1* reduces cellular autophagy and apoptosis through the Akt/mTOR signaling pathway [34]. Furthermore, *ARRB1* increases the transcriptional activity and expression of β-catenin and Akt to promote cell migration [35]. Therefore, it is speculated that *ARRB1* may be an important TF during follicle development in sheep, but this finding requires further experimental validation.

In the ceRNA network of QE vs. HE groups, we observed two modes of regulation between DEcircRNAs, DEmiRNAs, and DEmRNAs. miR-29a/b had the most nodes in the regulatory network, and its expression was upregulated in the QE vs. HE groups, which targeted and showed a negative correlation with *CUEDC1* and *KPNB1*. The involvement of miR-29a in reproductive processes in female mammals has been reported [36,37]. miR-29a overexpression inhibits ovarian cell proliferation and the cell cycle and decreases aromatase secretion and estradiol production [38]. miR-29b expression is elevated during the luteal phase, which affects progesterone (PROG) secretion by decreasing oxytocin receptor (OXTR) expression [39]. *KPNB1* is an important factor in cell proliferation and the regulation of follicle growth and development [40,41]. *CUEDC1* enrichment during oocyte maturation [42] affects the transcriptional activity of the TGF-beta/*SMAD* signaling pathway [43]. In the present study, 6 circRNAs (novel_circ_0033045, novel_circ_0003094, novel_circ_0007224, nov-el_circ_0015526, novel_circ_0003851, and novel_circ_0011620,) as well as *CUEDC1* and *KPNB1* jointly targeted miR-29a/b and were significantly downregulated in the QE vs. HE groups. It is speculated that these circRNAs release inhibitory factors on the target genes of *CUEDC1* and *KPNB1* by adsorbing miR-29a/b, thereby affecting sheep follicle growth and development. miR-17-5p had multiple nodes in our PPI network. miR-17-5P is associated with ovarian function [44], and the upregulation of miR-17-5p promotes follicle development marker gene (*LHR*, *CYP19A1*, and *AREG*) expression and estradiol synthesis [45]. In our study, novel_circ_0008117, novel_circ_0032347, novel_circ_0027336, and *ZFPM2* were significantly upregulated and jointly targeted the downregulation of miR-17-5P in the QE vs. HE groups. *ZFPM2* encodes a protein that affects gonadal development and plays an important role in follicular development [46,47] and hormone secretion [48]. Thus, it is speculated that nov-el_circ_0008117/novel_circ_0032347/novel_circ_0027336 might act as a ceRNA sponge for miR-17-5p and affect *ZFPM2* expression. Therefore, the DEcircRNAs-DEmiRNAs-DEmRNA regulatory network constructed in this study might affect steroid hormone synthesis, granulosa cell proliferation, and apoptosis, thereby influencing the follicular development in sheep.

This study had some limitations. Although RNA pull-down experiments revealed that some circRNAs and miRNAs are correlated with ovarian function in sheep, further in vitro and in vivo experiments are needed to confirm their expression and functional mechanisms.

## 5. Conclusions

In this study, we established the expression profile of circRNAs in the ovarian tissues of Cele black sheep and Hetian sheep and identified 180 differential circRNAs. Construction of ceRNA networks indicated that some functional circRNAs acted as follicle growth- and development-related miRNAs (miR-29a, miR-29b, miR-17-5p) by sponging functional miRNA mRNAs (*CUEDC1*, *KPNB1*, *ZFPM2*) of ceRNAs. Therefore, our findings provide new insights into sheep follicle growth and developmental mechanisms.

## Figures and Tables

**Figure 1 animals-13-00236-f001:**
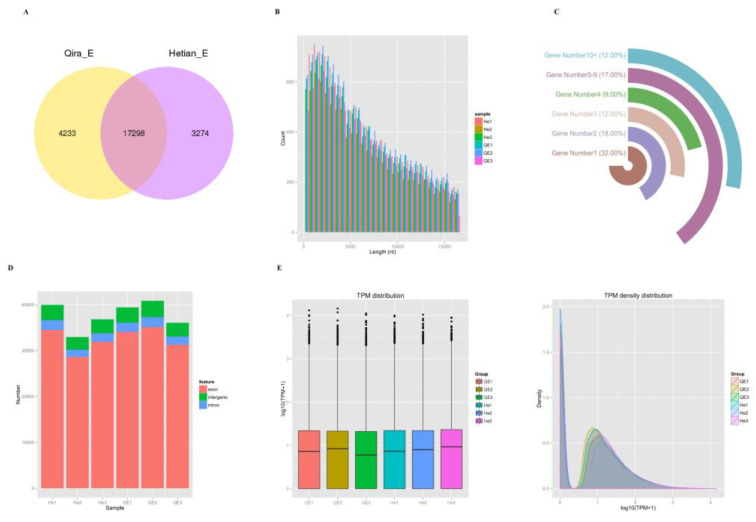
General characteristics of circRNAs in sheep ovaries. (**A**) Venn diagram of circRNA. (**B**) Length distribution of circRNA in sheep ovaries. (**C**) Distribution of circRNA host genes. More than 30% of host genes (1608) produce one circRNA, 918 host genes produce two circRNAs, and 506 host genes produce more than 10 circRNAs. (**D**) Distribution of circRNA genomic regions. (**E**) The expression of circRNAs in each sample was counted and normalized using TPM.

**Figure 2 animals-13-00236-f002:**
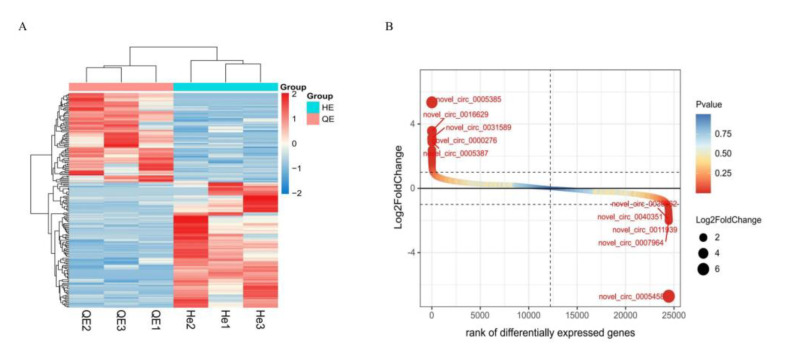
Differentially expressed circRNAs. (**A**) Heat map of DEcircRNAs. Each column shows one sample and each row represents one circRNA. (**B**) Bubble-rank plot of differential gene expression of RNA-seq.

**Figure 3 animals-13-00236-f003:**
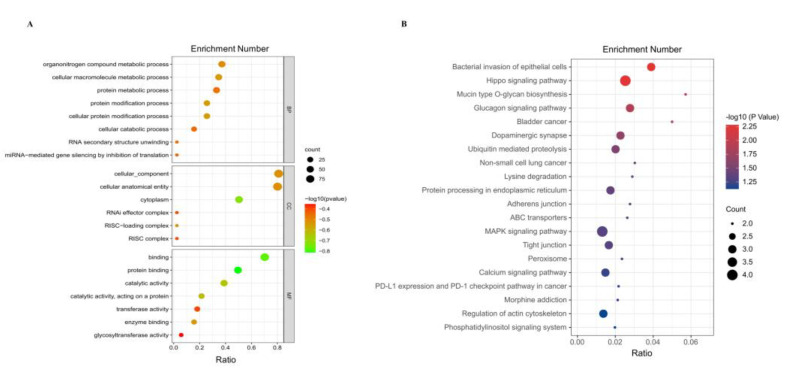
GO and KEGG enrichment analysis of DEcircRNA source genes. (**A**) GO analysis of DEcircRNA host genes in QE vs HE. The results are summarized in three main categories, biological processes (BP), cellular components (CC) and molecular functions (MF). (**B**) KEGG analysis of DEcircRNA host genes in QE vs HE.

**Figure 4 animals-13-00236-f004:**
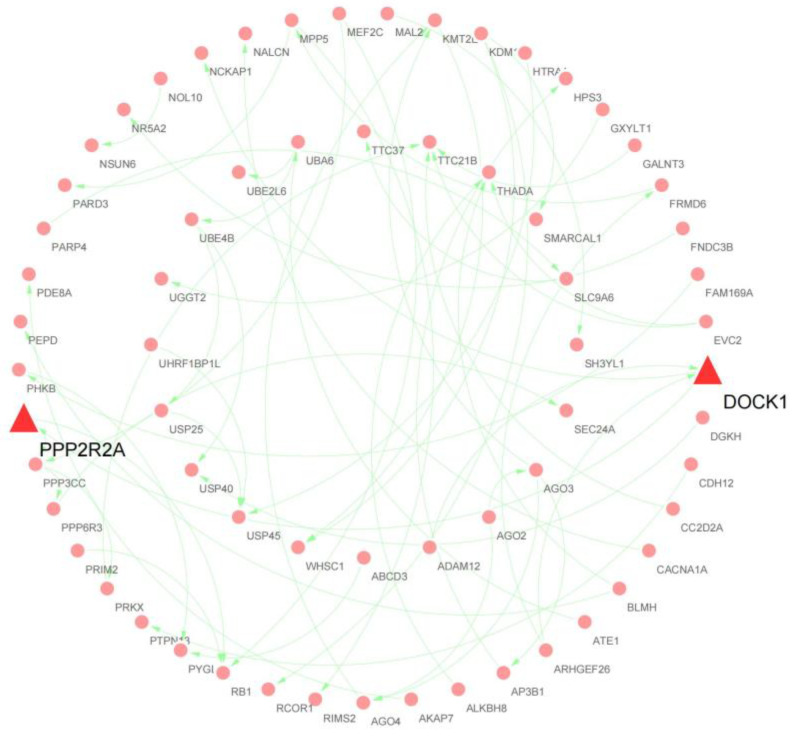
Analysis of the DEcircRNA host gene PPI network. The red and pink colors represent the identified host genes participating in the network; the central host genes are indicated by red triangles.

**Figure 5 animals-13-00236-f005:**
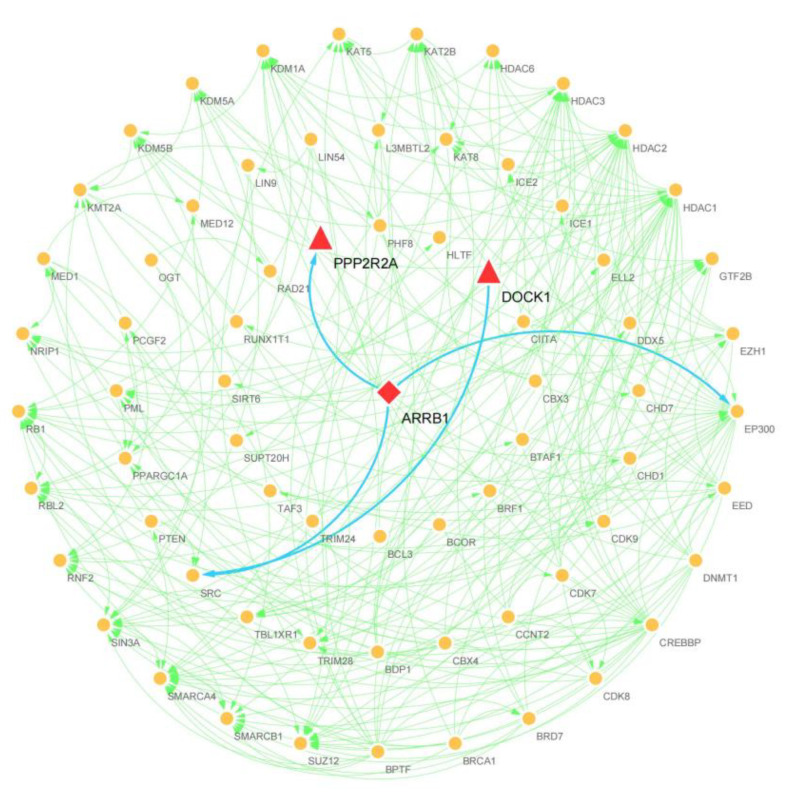
Regulatory network of central host genes and TFs of DEcircRNAs constructed based on Cytoscape software. Transcription factors are indicated by circles; predicted key transcription factors are indicated by diamonds; triangles represent central host genes.

**Figure 6 animals-13-00236-f006:**
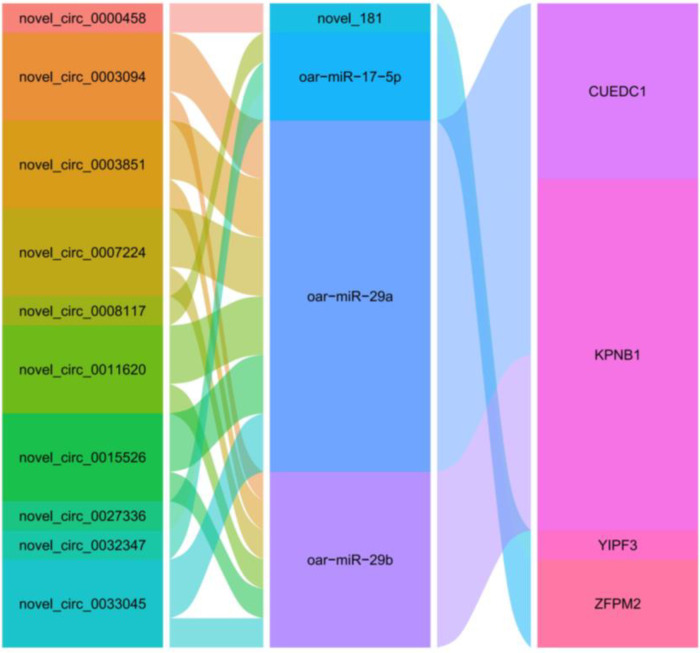
Co-expression network of DEcircRNA-DEDEmiRNA-mRNA. The left column represents DEcircRNA, the middle column represents DEmiRNA, the right represents DEmRNA, and the edges represent the relationship between them. The larger edge widths indicate the number of pathway connections.

**Figure 7 animals-13-00236-f007:**
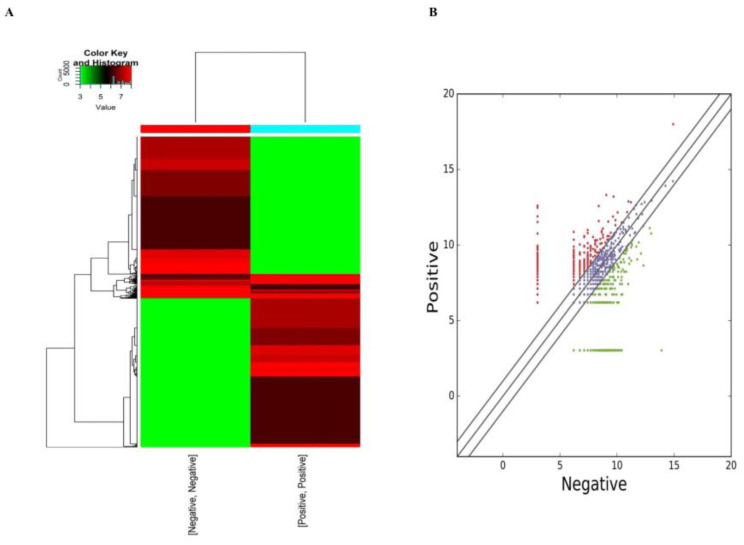
Oar-miRNA-125b potentially binding DEcircRNA. (**A**) Heat map of 3040 DEcircRNA. Each column shows one sample and each row represents one circRNA. (**B**) Positive vs. negative DEcircRNA scatter plots. Red dots indicate upward adjustment, turquoise dots indicate downward adjustment.

**Figure 8 animals-13-00236-f008:**
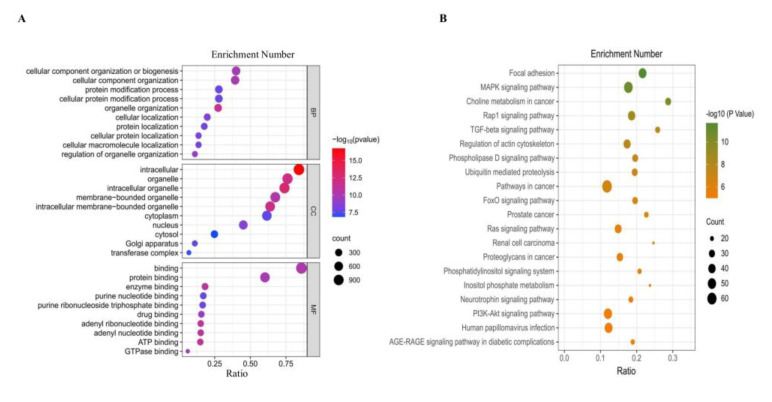
GO and KEGG enrichment analysis of DEcircRNA host genes that may be bound by oar-miRNA-125b (**A**) GO analysis of DEcircRNA host genes in positive vs. negative. The results are summarized in three main categories, BP, CC and MF. (**B**) KEGG analysis of DEcircRNA host genes in positive vs. negative.

**Figure 9 animals-13-00236-f009:**
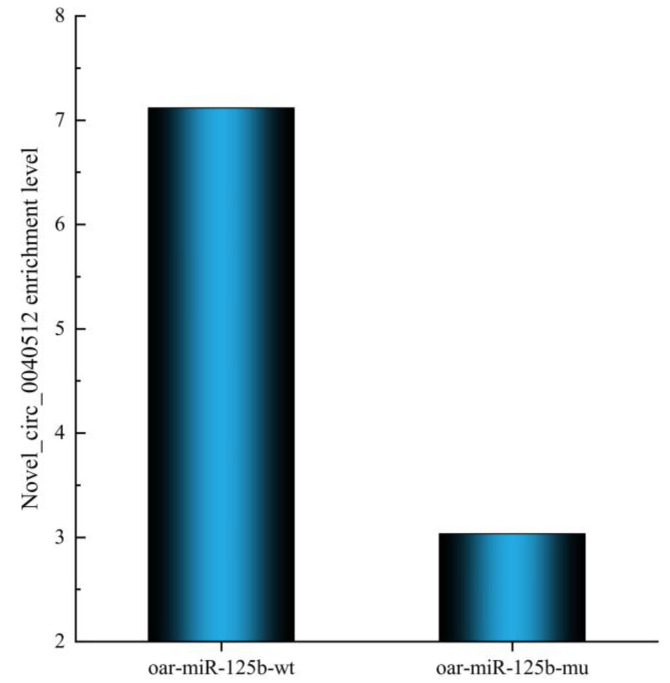
RNA pull-down assay to confirm the target relationship between novel-circ-0040512 and oar-miR-125b in cells.

**Table 1 animals-13-00236-t001:** Quality summary of sequencing data.

Sample Name	Raw Reads	Clean Reads	Error Rate (%)	Q20 (%)	Q30 (%)	GC Content (%)
QE1	78,346,928	76,379,904	0.03	94.93	88.52	61.88
QE2	69,340,482	67,725,550	0.03	95.06	88.78	61.16
QE3	74,602,968	73,065,816	0.04	94.67	88.03	62.52
HE1	79,269,182	77,666,732	0.03	94.94	88.51	62.62
HE2	75,472,380	73,136,040	0.03	95.6	89.99	62.7
HE3	72,610,104	70,164,462	0.03	95.81	90.38	61.35

## Data Availability

This can be addressed to the corresponding author.

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
