# Peer review of "Comprehensive Analysis of Differentially Expressed CircRNAs in the Ovaries of Low- and High-Fertility Sheep"

_animals, 2023, doi:10.3390/ani13020236_

Round 1

Reviewer 1 Report

In the present study, the authors performed transcriptome sequencing for Cele black sheep and Hetian sheep and identified circRNAs. Through target prediction via database, they further constructed a DEcircRNA-DEmiRNA-DEmRNA network. In the lase part, they focused on miR-125b and obtained potential binding DEcircRNAs. The authors aimed to clarify the molecular biology for the genetic progression of reproductive traits in sheep.

Even if the article concentrates on the hot topic categories of the reproductive field research, it lacks adequate evidence to draw a conclusion in the current form, mainly due to some pivotal mistake and disconnected study design in the manuscript. For this reason cannot be accepted for publication in the current form, in my opinion. Please, see the main points below:

1. Study design. The study could be divided into two parts. In the first part, a ceRNA network was finally constructed; in the second part, the novel-circ-0040512/miR-125b relationship was established. However, one of the most concerning issues for me regards the disconnection between these two parts. Neither novel-circ-0040512 nor miR-125b existed in Figure 6, so are they of importance for further investigation.

2. The ceRNA networks. In method section “2.9. Construction of DEcircRNA-DEmiRNA-DEmRNA networks”, the criteria of included miRNAs and mRNAs should be fully elucidated. For example, what is the minimum databases required to identify these RNAs (commonly predicted by at least 1/2/3 databases)? What are the databases and methods to obtain the mRNAs within ceRNA network? In addition, whether the miRNAs and mRNAs were compared with the sequencing data? The fold changes of mRNAs and miRNAs have been described in Supplementary Table S6, but relevant content can not be found in methods or results sections.

In addition to their paired relation, the content of ceRNA should also be considered. That is to say, the increase of miRNAs should be accompanied by the decrease of both circRNAs and mRNAs. However, some ceRNAs in the manuscript did not meet the requirements, for instance, the novel_circ_0008546/oar-miR-29a/CUEDC1 axis.

It is better to perform compare the alteration using qRT-PCR method.

3. Accurate writing. More details should be added in the method section, especially in “2.6. Differential expression analysis”. As mentioned above, add the criteria for DEcircRNA, DEmiRNA, and DEmRNA, and their fold change levels. Add the criteria for DEcircRNA in RNA pull-down assay. In line 216, P-value < 0.05 is inconsistent with “adjusted P-value” in line 130.

In Figure 2A, the group color and text are srong and misleading.

In Figure 2B, 3A and 8A, they lack interpretations for the coordinate axis.

Minor

1. English editing and scientific language: Many unnecessary abbreviations (e.g., GO in line 135, 224, 226, 296), unexplained ones that came out for the first time (e.g., line 19 MS).

2. Title. Replace “ two different sheep species” with “low- and high-fertility sheep”.

3. Line 403 and 418, sheep.

4. What is Wada sheep in line 16?

5. What is the difference between red and pink colors in Figure 4?

Based on the previous comments, I cannot recommend this paper for publication in the current form.

Author Response

Dear Reviewers:

Thank you for the reviewers’ comments concerning our manuscript entitled “Comprehensive analysis of differentially expressed circRNAs and their ceRNA networks in the ovaries of two different sheep species” (ID: animals-2094525). Those comments are valuable and very helpful for revising and improving our paper, as well as the important guiding significance to our researches. We have studied comment carefully and have made correction which we hope meet with approval.

We apologize for the confusion generated by the previous version of the manuscript and sincerely hope that our logic is now easier to follow with this new version. We worked on the manuscript for a long time and the repeated addition and removal of sentences and sections obviously led to poor readability. We have now worked on both language and readability and have also involved professional English speakers for language corrections. We really hope that the flow and language level have been substantially improved.

Point 1: Study design. The study could be divided into two parts. In the first part, a ceRNA network was finally constructed; in the second part, the novel-circ-0040512/miR-125b relationship was established. However, one of the most concerning issues for me regards the disconnection between these two parts. Neither novel-circ-0040512 nor miR-125b existed in Figure 6, so are they of importance for further investigation.

Response 1: Thank you very much for your valuable comments.The problem of disconnection between miRNA-125b and novel-circ-0040512, targeting the relationship and construction of the ceRNA network, is described in this article. In previous studies conducted by the project team, miRNA-125b was found to be differentially expressed in the ovaries of two breeds of sheep during estrus. A review the relevant literature identified the STAT3 gene as the miRNA-125b target gene, and further experiments demonstrated that miRNA-125b expression could affect STAT3 gene function. In this experiment, 11 DEcircRNAs targeting miRNA-125b were found through the DEcircRNA-DEmiRNA network; unfortunately, no miRNA-125b targeting DEmRNA was obtained. Ultimately, this resulted in the construction of the ceRNA network with no DEcircRNA/miRNA-125 b/DE mRNA axis, thus, causing incoherent visualization of the experiment. Previous studies have shown that some circRNAs can function through the miRNA-125b/STAT3 axis. In addition, in the article "3. Results" (3.6 Construction of a DEcircRNA-DEmiRNA-DEmRNA regulatory network) we have added a small part to improve the reader's understanding. Therefore, thank you very much for your question on further deepening the study of miRNA-125b with novel-circ-0040512, which is very important for our future experimental research and thanks again for your helpful suggestions.

Point 2: The ceRNA networks. In method section “2.9. Construction of DEcircRNA-DEmiRNA-DEmRNA networks”, the criteria of included miRNAs and mRNAs should be fully elucidated. For example, what is the minimum databases required to identify these RNAs (commonly predicted by at least 1/2/3 databases)? What are the databases and methods to obtain the mRNAs within ceRNA network? In addition, whether the miRNAs and mRNAs were compared with the sequencing data? The fold changes of mRNAs and miRNAs have been described in Supplementary Table S6, but relevant content can not be found in methods or results sections.

Response 2: Thanks for your advice. We apologize for the confusion generated by the previous version of the manuscript and sincerely hope that our logic is now easier to follow with this new version. We are very sorry again about our carelessness writing, and we have modified in article " 2. Materials and methods " (2.9. Construction of DEcircRNA-DEmiRNA-DEmRNA networks).

Point 3: In addition to their paired relation, the content of ceRNA should also be considered. That is to say, the increase of miRNAs should be accompanied by the decrease of both circRNAs and mRNAs. However, some ceRNAs in the manuscript did not meet the requirements, for instance, the novel_circ_0008546/oar-miR-29a/CUEDC1 axis.

Response 3: Thank you again for your valuable comments. We are happy to edit the text further based on the helpful comments of the reviewers and have already revised it in the article " 3. Results " (3.6. Construction of a DEcircRNA-DEmiRNA-DEmRNA regulatory network).

Point 4: It is better to perform compare the alteration using qRT-PCR method.

Response 4: Thank you very much for your valuable comments, your suggestions have helped us a lot. I understand that qRT-PCR can improve the integrity of the experiment, which helps us a lot in our experimental studies, and we really want to improve the whole experiment substantially. However, in the preliminary study of related experiments and sequencing results, we have basically completed the screening of circRNAs that may be involved in influencing follicular development in sheep. Next, we will further deepen the study on the mechanism of circRNA affecting sheep follicle development more accurately and rigorously by qRT-PCR, gene knockdown, gene overexpression and dual luciferase gene detection in our future circRNA function and mechanism studies. I feel honored to receive your valuable comments and thank you again!

Point 5: Accurate writing. More details should be added in the method section, especially in “2.6. Differential expression analysis”. As mentioned above, add the criteria for DEcircRNA, DEmiRNA, and DEmRNA, and their fold change levels. Add the criteria for DEcircRNA in RNA pull-down assay. In line 216, “P-value < 0.05” is inconsistent with “adjusted P-value” in line 130.

Response 5: We will be happy to edit the text further, based on helpful comments from the reviewers. We are very sorry again about our carelessness writing and we have modified in article.

Point 6: In Figure 2A, the group color and text are srong and misleading.

Response 6: Thanks for your advice. We are very sorry again about our carelessness, and we have modified in manuscript as appropriate.

Point 7: In Figure 2B, 3A and 8A, they lack interpretations for the coordinate axis.

Response 7: Thank you very much for your valuable comments. We will be happy to edit the text further, based on helpful comments from the reviewers.

Point 8: English editing and scientific language: Many unnecessary abbreviations (e.g., GO in line 135, 224, 226, 296), unexplained ones that came out for the first time (e.g., line 19 “MS”).

Response 8Thanks for your advice. We are very sorry again about our carelessness writing, and we have modified in article.

Point 9: Title. Replace “two different sheep species” with “low- and high-fertility sheep”.

Response 9: We will be happy to edit the text further, based on helpful comments from the reviewers. We agree with this suggestion and have modified the manuscript as appropriate. Thanks again for your helpful suggestions.

Point 10: Line 403 and 418, “sheep”.

Response 10: Thanks for your advice. We are very sorry again about our carelessness, and we have modified in manuscript as appropriate.

Point 11: What is Wada sheep in line 16?

Response 11: Thank you very much for your suggestion. Again, we are very sorry for our carelessness and we have made changes in the article.

Point 12: What is the difference between red and pink colors in Figure 4?Response 12: Thank you very much for your valuable comments. In Figure 4, the analysis of circRNA host genes by PPI network combined with MCODE plug-in, the DOCK1, PPP2R2A genes are considered as key central host genes based on the maximum number of nodes and scores. The red triangles represent the central host genes (DOCK1, PPP2R2A), and the pink are other host genes. Apologies for the confusion we caused you by our careless writing, and thanks again for your valuable input!

We feel great thanks for your professional review work on our article. Please refer to the attachment for details.

Reviewer 2 Report

Major issues:

1> Some knockdown or overexpression experiments need to done to validate the results from sequencing

Minor issues:

1> Figure2b, texts should be adjusted to appropriate size

2> For figure 3, high resolution pictures should be used and texts are twisted

Author Response

Dear Reviewers:

Thank you for the reviewers’ comments concerning our manuscript entitled “Comprehensive analysis of differentially expressed circRNAs and their ceRNA networks in the ovaries of two different sheep species” (ID: animals-2094525). Those comments are valuable and very helpful for revising and improving our paper, as well as the important guiding significance to our researches. We have studied comment carefully and have made correction which we hope meet with approval.We apologize for the confusion generated by the previous version of the manuscript and sincerely hope that our logic is now easier to follow with this new version.

Point 1: Some knockdown or overexpression experiments need to done to validate the results from sequencing.

Response 1: It is a great honor to receive your valuable comments and your suggestions are very helpful to us. I understand that gene knockdown can improve the integrity of the experiment, which helps us a lot in our experimental studies, and we really want to improve the whole experiment substantially. However, in the preliminary study of related experiments and sequencing results, we have basically completed the screening of circRNAs that may be involved in influencing follicular development in sheep. Next, we will further deepen the study on the mechanism of circRNA affecting sheep follicle development more accurately and rigorously by qRT-PCR, gene knockdown, gene overexpression and dual luciferase gene detection in our future circRNA function and mechanism studies. Thank you very much again for your valuable comments!

Point 2: Figure2b, texts should be adjusted to appropriate size.

Response 2: Thank you for your advice. We will be happy to edit the text further, based on helpful comments from the reviewers.

Point 3: For figure 3, high resolution pictures should be used and texts are twisted.

Response 3: Thanks for your advice. We are very sorry again about our carelessness, and we have modified in figure as appropriate.

We feel great thanks for your professional review work on our article. Please refer to the attachment for details.

Reviewer 3 Report

The authors looked and differences in the expression of circRNA between two breeds of sheep with different genetic backgrounds and prolificacy to identify potential targets of circRNA involved in follicular growth and ovulation. The study is interesting but needs improvement before publication. Comments are below.

Please simplify the title: Analysis of differentially expressed circRNAs-ceRNA networks in the ovaries of two sheep species

Simple summary:

please check English. Avoid using the same word in the same sentence or the next sentence (ex. Sheep, line 11 and 12; identified (line 12 and 13). Also, typo mistakes (ex. line 19).

Not clear what the authors mean by “perennially estrous”. Please find another word

                The sentence “Identification of key circRNAs and miRNAs, including miR-29a, miR-29b and 17 miR-17-5p, by bioinformatics analysis”. Has no connection with the text.

Abstract: needs English revision and coherence improvement. competing endogenous RNA ceRNAs is not properly introduce.

Introduction: English improvement need it. The text just doesn’t make sense. Sentences should strat with a number in “number” format.

M&M:

Please include the number of animals (I understand were 3 animals per group Q1.. and so on? ), ovaries, and how estrous synchronization was established, how many times per day was estrous detected? Was it at the same time? It is not clear whether the conditions of the animals (housing, nutrition,) were the same for all the animals. Please include more information regarding age and weight of the animals used for the experiment.

Were ovulated animals mixed with pre-ovulated animals?

It is not clear how many biological replicates are in this study.

Results:

QE HE is not previously defined.  Please prove the results per ovary rather than per animal (Q1a, Q1b suggested).

Figure 5 is unreadable.  

Please discuss the potential reasons for differences in the amount of circRNA found in your studied compared to others. Also, the uniquely expressed circRNA are not specie specific but breed related. (line 347).

The discussion needs English revision and to be more focused on the differences between the two genetic backgrounds of sheep used and the differences found. Line 351: in our another study??/ This is not clear and the study is not cited.

The last paragraph of the discussion is long and should be shortened. Please don’t use a whole paragraph for the limitation of the study.

ARRB1 hypothesis should be in the discussion (does not belong to the experiment of this study,).

Line 339: what previous studies?

Several circRNA can be produced by a gene that is not novel. Please cite and discuss your finding. Why is this finding important in this research?

Author Response

Dear Reviewers:

Thank you for the reviewers’ comments concerning our manuscript entitled “Comprehensive analysis of differentially expressed circRNAs and their ceRNA networks in the ovaries of two different sheep species” (ID: animals-2094525). Those comments are valuable and very helpful for revising and improving our paper, as well as the important guiding significance to our researches. We have studied comment carefully and have made correction which we hope meet with approval.

Point 1: Please simplify the title: Analysis of differentially expressed circRNAs-ceRNA networks in the ovaries of two sheep species

Response 1: Thank you very much for your valuable comments! We will be happy to edit the text further, based on helpful comments from the reviewers. Thanks again for your comments.

Point 2: please check English. Avoid using the same word in the same sentence or the next sentence (ex. Sheep, line 11 and 12; identified (line 12 and 13). Also, typo mistakes (ex. line 19).

Response 2: Thanks for your advice. We are very sorry again about our carelessness writing, and we have modified in article.

Point 3: Not clear what the authors mean by “perennially estrous”. Please find another word.

Response 3: We are very sorry again about our carelessness writing and we have modified in article.

Point 4: The sentence “Identification of key circRNAs and miRNAs, including miR-29a, miR-29b and 17 miR-17-5p, by bioinformatics analysis”. Has no connection with the text.

Response 4: Thanks for your advice. We will be happy to edit the text further, based on helpful comments from the reviewers.

Point 5: Abstract: needs English revision and coherence improvement. competing endogenous RNA ceRNAs is not properly introduce.

Response 5: We apologize for the poor language of our manuscript. We worked on the manuscript for a long time and the repeated addition and removal of sentences and sections obviously led to poor readability. We have now worked on both language and readability and have also involved professional English speakers for language corrections. We really hope that the flow and language level have been substantially improved.

Point 6: Please include the number of animals (I understand were 3 animals per group Q1.. and so on? ), ovaries, and how estrous synchronization was established, how many times per day was estrous detected? Was it at the same time? It is not clear whether the conditions of the animals (housing, nutrition,) were the same for all the animals. Please include more information regarding age and weight of the animals used for the experiment.

Response 6: Thank you again for your valuable comments. We are happy to edit the text further based on the helpful comments of the reviewers and have already revised it in the article " 2. Materials and methods " (2.2. Laboratory animal samples and collection).

Point 7: Were ovulated animals mixed with pre-ovulated animals?It is not clear how many biological replicates are in this study.

Response 7: Thanks for your advice. We are very sorry again about our carelessness, and We are happy to edit the text further based on the helpful comments of the reviewers and have already revised it in the article " 2. Materials and methods " (2.2. Laboratory animal samples and collection).

Point 8: QE HE is not previously defined.  Please prove the results per ovary rather than per animal (Q1a, Q1b suggested).

Response 8: Thanks for your advice. We apologize for the confusion generated by the previous version of the manuscript and sincerely hope that our logic is now easier to follow with this new version. We are very sorry again about our carelessness writing, and we have modified in article " 2. Materials and methods " (2.2. Laboratory animal samples and collection).

Point 9: Figure 5 is unreadable.

Response 9: We are very sorry again about our carelessness writing and we have modified in article.

Point 10: Please discuss the potential reasons for differences in the amount of circRNA found in your studied compared to others. Also, the uniquely expressed circRNA are not specie specific but breed related. (line 347).

Response 10: Again, thank you very much for your valuable comments. It is believed that the reasons for the different circRNA amounts are caused by including tissue specificity and developmental different stage specificity and breed specificity. We have modified in article, and will be happy to edit the text further, based on helpful comments from the reviewers.

Point 11: The discussion needs English revision and to be more focused on the differences between the two genetic backgrounds of sheep used and the differences found. Line 351: in our another study??/ This is not clear and the study is not cited.

Response 11: Again, we are very sorry for our careless writing, and we have invited an English professional to revise the article, which we believe will substantially improve the content reading and comprehension. In addition, the relevant literature has been cited and revised "in our study". Again, we thank you for your important suggestions.

Point 12: The last paragraph of the discussion is long and should be shortened. Please don’t use a whole paragraph for the limitation of the study.

Response 12: Thanks for your advice. We are very sorry again about our carelessness writing, and we have modified in article.

Point 13: ARRB1 hypothesis should be in the discussion (does not belong to the experiment of this study,).

Response 13: Thank you for your suggestion. Again, we are very sorry for our careless writing and strongly agree with your suggestion and have made changes in the article.

Point 14: Line 339: what previous studies?

Response 14: Thanks for your advice. We are very sorry again about our carelessness, and we have modified in manuscript as appropriate.

Point 15: Several circRNA can be produced by a gene that is not novel. Please cite and discuss your finding. Why is this finding important in this research?

Response 15: Thank you again for your suggestion. As it has been shown in past studies that circRNAs derive from the same transcriptional product as host gene mRNAs and are able to regulate host gene transcription and expression, it was speculated whether circRNAs could directly regulate functions related to reproduction-related genes, hence the preliminary discussion. Again, we are very sorry for our careless writing, and we have made changes in the article.

We feel great thanks for your professional review work on our article. Please refer to the attachment for details.

Round 2

Reviewer 1 Report

Thank you for the authors and the current manuscript has been greatly improved.

My questions have been addressed, while some minor issues still need to be corrected.

Line 37: circRNA. "ceRNA" is recommended to be added.

Line 147:  transcription factor (TF)

Line 165: DEmRNAs

Table 1: HE1, HE2, HE3

Figure 7: The meanings of red and turquoise.

Line 323: 3,040

Author Response

Dear Reviewers:

We feel great thanks for your professional review work on article. According to your nice suggestions, we have made extensive corrections to our previous draft, the detailed corrections are listed below.

Point:

Line 37: circRNA. "ceRNA" is recommended to be added.

Line 147:  transcription factor (TF).

Line 165: DEmRNAs.

Table 1: HE1, HE2, HE3.

Figure 7: The meanings of red and turquoise.

Line 323: 3,040.

Response: I am very sorry for our mistakes. We have made changes to each of these issues in the manuscript, as you are concerned about. Thank you again for your corrections.
